# Axin Family of Scaffolding Proteins in Development: Lessons from *C. elegans*

**DOI:** 10.3390/jdb7040020

**Published:** 2019-10-15

**Authors:** Avijit Mallick, Shane K. B. Taylor, Ayush Ranawade, Bhagwati P. Gupta

**Affiliations:** 1Department of Biology, McMaster University, Hamilton, ON L8S-4K1, Canada; mallia1@mcmaster.ca (A.M.); taylos49@mcmaster.ca (S.K.B.T.); 2Department of Bioengineering, Northeastern University, Boston, MA 02115, USA; ayushranawade@gmail.com

**Keywords:** Axin, *C. elegans*, *pry-1*, *axl-1*, WNT signaling, scaffolding protein, signal transduction, development

## Abstract

Scaffold proteins serve important roles in cellular signaling by integrating inputs from multiple signaling molecules to regulate downstream effectors that, in turn, carry out specific biological functions. One such protein, Axin, represents a major evolutionarily conserved scaffold protein in metazoans that participates in the WNT pathway and other pathways to regulate diverse cellular processes. This review summarizes the vast amount of literature on the regulation and functions of the Axin family of genes in eukaryotes, with a specific focus on *Caenorhabditis elegans* development. By combining early studies with recent findings, the review is aimed to serve as an updated reference for the roles of Axin in *C. elegans* and other model systems.

Axin was first discovered as a negative regulator of WNT (*wingless* and *int-1*) signaling in mice while deciphering its role in embryonic axis formation [1]. This protein is the product of the mouse Fu (Fused) gene [2,3,4] and was named Axin for its initial discovered role in inhibiting axis formation (Axis inhibition). Soon after, Axin and its homolog Axil (for Axin-like, also known as Axin2 or conductin) were discovered in a yeast two-hybrid screen for GSK-3β (glycogen synthase kinase-3 beta)-interacting proteins [5,6]. Since then, involvement of Axin family proteins in WNT signaling has been extensively characterized, revealing that they can bind to and facilitate interactions between several WNT pathway components such as the WNT co-receptor LRP (low-density lipoprotein-related protein), Dvl (Dishevelled), APC (tumor suppressor Adenomatous Polyposis Coli), GSK-3β, β-catenin, CKs (casein kinases), and many others [7,8,9,10,11,12,13,14,15,16,17,18].

Over the years, Axin homologs have been identified in several organisms. Many of these discoveries were facilitated by the availability of whole-genome sequences. While functional studies are currently limited to a few animal models, the findings and sequence data show that Axin is conserved in metazoans, with invertebrates carrying an ancestral gene and higher eukaryotes possessing two distinct *Axin1* and *Axin2* genes [1,5,19,20]. The nematode lineage contains an additional, divergent, Axin homolog [21] (see Figure 1 and further discussion below). Experiments performed in mice have revealed that while both Axin proteins regulate WNT signaling, they have different functions [22,23,24]. Whereas Axin1 is ubiquitously expressed in embryos and is essential for viability, Axin2 is restricted to a few tissues and serves as a transcriptional target of WNT signaling [25,26]. Similar to the published literature, the ‘Axin1’ and ‘Axin’ names have been used interchangeably in this article, whereas the term ‘Axin family’ refers to both Axin1 and Axin2 homologs that share conserved protein-interaction domains.

## 1. Axin Domains

Axin possesses multiple regions that facilitate its interactions with various proteins (Figure 2). One of these regions is involved in regulating G protein signaling (the RGS domain) near the N-terminus that can bind the APC protein [11] (Figure 2). In the context of the WNT pathway, APC requires Axin to form a destruction complex with GSK-3β and other proteins [27]. The C-terminus of Axin possess a DIX (Dishevelled/Axin homologous) domain that facilitates WNT pathway-specific interactions by forming homodimers and heterodimers with the Axin and Dvl proteins [14,28] (Figure 2). In addition to these well-defined domains, Axin also contains regions between the RGS and DIX domains that bind β-catenin (in Armadillo repeats 2–7) [5] and two serine/threonine kinases GSK-3β [29] and CKIα (casein kinase I) [30] (Figure 2). Through these regions, Axin recruits APC, GSK-3β, CKIα, and β-catenin to form a multimeric complex in the absence of WNT signaling. The complex causes enhanced phosphorylation of β-catenin by GSK-3β and CKIα [31,32] and targets it for ubiquitination and proteasomal degradation. Activation of WNT signaling inhibits the destruction of β-catenin and promotes nuclear translocation of the non-phosphorylated form and activation of WNT-responsive genes [32]. Axin also possesses additional sequences that can facilitate protein–protein interactions, thereby promoting activation of other, non-WNT pathway, factors ([33], also see below).

## 2. Overview of the Developmental Roles of the Axin Family

### 2.1. Vertebrate Models

The Axin family of genes are involved in diverse developmental processes. Analyses of mutant and gene knock-down experiments in different animal models have shown defects in anterior–posterior-axis formation and organogenesis. In some cases, these abnormalities may also contribute to early stage lethality. As mentioned above, Axin was initially identified for its role in embryonic axis formation, as mutant mice exhibited axial defects [1]. Subsequently, the gene was also shown to be essential for viability and the formation of many other organs, including the heart, tail, primitive streak, brain, and muscles [24,34,35,36]. In zebrafish, mutations in Axin cause abnormal fate determination of the eyes and telencephalon, with defective establishment of asymmetries of the nervous system [37,38]. Experiments in *Xenopus laevis* revealed that a failure to regulate Axin activity results in duplication of the dorsal axis due to the constitutive activation of WNT signaling as determined by the expression analysis of target genes [39].

In addition to their requirements for the formation of axis and organs, Axin family members also play crucial roles in neuronal development. Alterations in Axin expression, caused by mutations, knock-down, or dysregulation, show defects in various processes including neurogenesis, neuronal differentiation, axon outgrowth, and synapse formation. As a scaffolding protein, Axin facilitates the recruitment of various proteins to regulate gene expression and cytoskeletal dynamics. Through these actions, Axin affects signaling pathway activities, e.g., WNT-β-catenin, JNK (c-Jun N-terminal kinase) and TGF-β (transforming growth factor-beta) (see below for more details), and organization of proteins such as microtubules (reviewed in [40,41]). Previously, it was shown that ectopic Axin expression in cultured cells blocked neuronal differentiation, a process that involved WNT-3a-β-catenin signaling [42]. In a separate study, it was demonstrated that Axin inhibition in a neuroblastoma cell line, through the application of Li (lithium) and a GSK-3β inhibitor, promoted neurite outgrowth, whereas ectopic Axin expression caused an opposite phenotype [43]. Subsequently, several reports have shown Axin’s role in neuronal proliferation and differentiation, axon formation, dendritic spine morphology, and synapse formation [44,45,46,47,48].

Cell proliferation is another process where Axin’s involvement has been investigated in considerable detail. The findings have shown that, as a negative regulator of the WNT-β-catenin signaling, Axin functions to inhibit cancerous growth and appears to act as a tumor suppressor [49]. Altered Axin regulation and activity are associated with various types of cancers such as lung cancer, colorectal cancer, and HCC (hepatocellular carcinoma) [50,51,52]. Characterizations of human HCC cultures identified *Axin1* mutations in many of the cell lines and a corresponding increase in the DNA-binding activities of TCF/LEF (T-cell factor/Lymphoid enhancer-binding factor 1) and β-catenin [53]. Interestingly, when human and mouse HCCs lacking Axin were examined, it was found that in most cases, human HCCs were not associated with increased β-catenin activation [54]. Moreover, HCC induction in mice due to *Axin1* mutation was independent of WNT-β-catenin signaling [54,55]. Further investigations on gene signatures of human and mouse HCCs revealed a significant overlap between genes affected by Axin1, Notch and YAP (Yes-associated protein), which may provide new avenues for treatments of Axin1-linked tumors [54].

Among its other roles, Axin appears to be necessary for maintaining cell survival, metabolic homeostasis, and thymic adipogenesis. Overexpression studies in transgenic mice and certain cultured cells demonstrated increased apoptosis, possibly due to activation of the cell-death pathway [56,57]. Axin can also regulate the activation of AMPK (AMP (adenosine monophosphate)-activated protein kinase), a sensor of the cellular-energy status [58]. Axin forms a complex with AMPK and a serine-threonine kinase LKB1 (Liver kinase B1), which then leads to AMPK activation to protect cells against increased stress, such as under a condition of low nutrient levels (see below for more discussion on this topic). Finally, in the case of thymus function, Axin promoted age-related adipogenic programming of thymic stromal cells [59]. This process is linked to reduced T-cell production and thymic involution, suggesting that any potential therapeutic intervention to prolong aging may involve lowering Axin activity.

### 2.2. Invertebrate Models

Studies in invertebrates have been instrumental in uncovering the developmental roles of Axin homologs in tissues and organs, in the context of intact animals. The *Drosophila melanogaster* Axin (D-Axin) homolog is necessary for the development of embryos and organs, such as the wings, eyes, heart, gut, and circulatory system [20,60,61,62]. Analysis of the *Tribolium castaneum* Axin homolog (Tc-Axin) revealed its dynamic expression during embryogenesis. Tc-Axin was initially localized at the anterior pole, extending posteriorly during subsequent development and eventually becoming somewhat ubiquitous [63]. This expression pattern was essential for the formation of head structures, considering that Tc-Axin knockdown led to an absence of the head and thoracic parts. In the case of the nematode *C. elegans*, two divergent Axin-like proteins, specifically PRY-1 (poly ray 1) and AXL-1 (Axin-like 1) have been identified. The roles of both these family members are discussed in a separate section.

In summary, the roles of Axin family of proteins described above rely on its scaffolding properties that are mediated by conserved domains facilitating interactions with WNT-β-catenin pathway components (Figure 2). Additionally, Axins utilize other unique regions (not shown in Figure 2) to recruit non-WNT pathway components involved in other developmental processes that are summarized in the next section (also see Figure 3).

## 3. Axin Proteins Interact with Many Factors Including Signaling Pathway Components

Given that Axins play essential roles in metazoan development, it is not surprising that Axin family members form complexes with various cellular factors and components of signal transduction pathways (Figure 2 and Figure 3A). While much research has focused on their involvement in the canonical WNT-β-catenin signaling pathway, which was summarized in the previous section, additional studies have demonstrated Axin’s participation in non-canonical WNT signaling (Figure 3A). These include the PCP (planar cell polarity) pathway that provides directional information during organ formation, the WNT/calcium pathway that regulates muscle contraction and PKC (protein kinase C) enzyme activation, and the Ror2 (receptor tyrosine kinase-like orphan receptor)- and Ryk (related to tyrosine kinase)-dependent WNT pathway in coordinating cell movement and polarity (Figure 3A) (reviewed in [64]). These WNT pathways are unique in that they do not utilize the canonical effector, β-catenin.

In addition to the above conserved WNT-mediated signaling events, studies in *C. elegans* have shown the presence of a divergent asymmetry pathway that regulates nuclear factor POP-1 in a WRM-1 (worm armadillo 1)/β-catenin-LIT-1 (loss of intestine 1)/NLK (Nemo-like kinase)-dependent manner (see [65] and references therein, [66]). In these cases, PRY-1 affects asymmetric POP-1 localization to control EMS (endomesodermal) precursor division in the embryo and seam cell divisions in larvae (discussed below).

Axin family members also cooperate with an increasing number of proteins in other, non-WNT, processes. These interactions involve pathways such as JNK, TGF-β, p53, and AMPK. Tissue culture experiments involving kidney 293T cells and embryonic fibroblast cells showed that Axin binds to MEKK1 and MEKK4, two members of the MEKK (MAPK (Mitogen-activated protein kinase)/ERK (Extracellular signal-regulated kinase) kinase kinase) family, through domains distinct from those involved in WNT signaling and activates the MKK4- and MKK7- (also belonging to MEKK family) mediated JNK cascade [67,68]. This Axin-dependent JNK activation is inhibited by the WNT pathway components Dvl, GSK-3β, CKIα, and CKIε. Furthermore, during dorsalization of zebrafish embryos, an Axin-interacting protein, Aida, inhibits Axin-mediated JNK activation by disrupting Axin homodimerization [69]. JNK signaling is a key regulator of various cellular processes occurring in response to external signals. Upon activation, JNK translocates to the nucleus and activates gene-expression changes.

Axin’s function in TGF-β pathway involves regulation of transcription factor Smad3 activity to affect gene transcription. In human MSCs (mesenchymal stem cells), Axin and GSK-3β physically interact to facilitate Smad3 phosphorylation by the active TβRI (TGF-β type-I receptor) kinase (reviewed in [70]). This interaction is needed to promote MSC proliferation. Axin and GSK-3β also act to regulate ubiquitin-dependent proteasomal degradation of Smad3 in human keratinocytes and hepatocellular carcinoma cells in a manner analogous to the β-catenin degradation process [71]. In yet another study, Axin acted as a scaffold to form a ternary complex with Smad7 and the ubiquitin E3 ligase Arkadia in cultured cells to enhance Sma7 ubiquitination, leading to the activation of TGF-β signaling [72].

While the tumor-suppressor role of Axin has been traditionally investigated in the context of WNT-β-catenin signaling, some studies have demonstrated its interactions with p53, a DNA-binding protein that responds to genotoxic stress and controls cell proliferation and cancerous growth. During p53 signaling, Axin interacts with HIPK2 (homeodomain-interacting protein kinase-2) to facilitate p53 phosphorylation, which stimulates p53-dependent transcription of target genes [73]. Subsequent work showed that this regulatory mechanism involves the formation of distinct complexes consisting of additional proteins such as Pirh2 (p53-induced RING-H2) and the histone acetyl transferase, Tip60 [74,75]. Axin can also associate with a death domain-associated protein, Daxx, to regulate p53 function to induce cell death following exposure to ultraviolet light [76].

Research on Axin has also uncovered its role in controlling cellular energy, nutrient sensing, and metabolic processes. One of these processes involves glucose homeostasis. In *Drosophila*, D-Axin was reported to physically interact with a component of the glucose-transport regulatory complex, DCAP (*Drosophila* catabolite activator protein), to increase glycogen utilization through insulin signaling and glucose transport [77,78]. Additionally, Axin formed a ternary complex with the TNKS2 (ADP (adenosine diphosphate)-ribosylase tankyrase 2) enzyme and the kinesin motor protein KIF3A in cultured cells to facilitate translocation of the insulin-stimulated glucose transporter, GLUT4, and glucose uptake [79]. In an unrelated study, mouse Axin2 was reported to participate in signaling through a *Drosophila* Pygo2 (Pygopus) homolog to affect glucose metabolism [80]. Another energy-homeostasis system involving Axin function is the AMPK signaling network. AMPK is a central player involved in sensing AMP and ADP levels in response to ATP (Adenosine triphosphate) consumption. During AMPK signaling, accumulation of AMP and glucose-starvation initiates Axin binding to LKB1 to enable AMPK phosphorylation [58]. Because Axin depletion inhibits AMPK stimulation, which results in the loss of lipid homeostasis, Axin acts as a metabolic rheostat and energy sensor [58].

Beyond their other functions, Axins also form complex with Dvl to facilitate cytoskeletal rearrangement during gastrulation (Reviewed in [34]) and orientation of the mitotic spindle during asymmetric cell division [81]. Both these proteins possess a common DIX domain that is responsible for this binding [82]. In addition, Axin interacts with other cellular components to regulate cytoskeletal arrangement. Cowan and Henkemeyer [83] showed that one of the ways that Axin participates in the process is by modulating Eph/ephrin-bidirectional signaling. Specifically, interactions of Axin with Grb4, a SH2/3 domain adaptor protein of the Eph/ephrin pathway, facilitates the recruitment of other proteins leading to cytoskeletal rearrangement during cell and axon growth–cone movement.

In summary, Axin participates in both WNT-dependent and -independent signaling events (Figure 3A). By acting as a scaffold protein, it helps recruit other factors to execute a wide variety of cellular and molecular processes in eukaryotes (Figure 3B). While Axin’s role in WNT-β-catenin signaling appears to be conserved in metazoans, several studies have also reported its involvement in other, WNT-independent, pathways. To what extent the later functions of Axin are conserved remains to be established, though it is worth pointing out that Axin-AMPK interaction has been shown in both mice and *C. elegans* systems (see the section ‘Regulation of Developmental Processes in *C. elegans*’).

## 4. Regulation of Axin Functions

The crucial roles of Axin family members in metazoans depend on multiple mechanisms that operate at spatiotemporal and subcellular levels. Several reports have shown that Axin is regulated both at transcriptional and post-translational levels. Using an auto-feedback loop, Axin controls its own expression [26,84,85,86], although the relevance of such a mechanism at the organismal and cellular levels remains to be understood. Modulation of protein functions can also occur via changes in their oligomeric state or through interactions with binding partners. As described above, Axin possesses a DIX domain at the C-terminus, which mediates both homo- and hetero-interactions, thus contributing to its essential activities [87,88,89,90]. The same domain is also found in other proteins, such as Dvl. It has been proposed that Dvl might recruit Axin from the destruction complex to the LRP receptor via DIX-domain interactions [91,92].

The post-translational modifications of Axin include phosphorylation, ubiquitination, and SUMOylation. These alterations affect the subcellular localization, stability, or potential interactions of the protein with other factors (reviewed in [93,94]). Similar to β-catenin, under basal conditions, Axin is phosphorylated by both GSK-3β and CKI [10,95,96] to function in the destruction complex and is subsequently dephosphorylated upon WNT pathway activation [10,97,98]. In updated regulation models, Axin phosphorylation occurs during both the ‘off’ and ‘on’ states of WNT signaling, and is dependent on another key destruction complex member, APC [99,100]. Another kinase that is reported to phosphorylate Axin is Cdk5 (Cyclin-dependent Kinase 5). During mouse cortex development, Cdk5-mediated phosphorylation was found to be necessary for the interaction of Axin with GSK-3β, leading to microtubules stabilization during axon formation [45].

Axin is subjected to ubiquitin-mediated proteolysis through poly ADP-ribose modification. Experiments in cell culture systems showed that poly ADP-ribosylation of Axin by ADP-ribose polymerase enzymes, tankyrase 1 and tankyrase 2, targeted Axin for proteasomal degradation when the pathway was in the ‘off’ state [101,102,103]. The role of tankyrase in Axin regulation has also been demonstrated in the *Drosophila* system [104,105]. Feng et al. [105] reported that Axin levels were moderately higher in Tnks (Tankyrase)-mutant flies. Since a further increase in Axin expression in animals lacking Tnks function disrupted expression of Wingless/WNT reporter gene, it was concluded that Tnks-dependent regulation normally acts to buffer Axin activity. The ubiquitination of Axin is facilitated by several ubiquitin E3 ligases. Ji et al. [106] used human cell lines to investigate the roles of two RING (really interesting new gene) family ligases, SIAH1 and SIAH2 (seven in absentia homologs 1 and 2), and found that Axin was ubiquitinated and degraded as a feed-forward mechanism to achieve sustained activity of WNT-β-catenin signaling. The *Drosophila* homolog, Iduna (also a RING family member), acts as a key factor in the breakdown of ADP-ribosylated Axin [62,107]. Other ligases (e.g., Smurf2 of HECT (homologous to the E6-AP carboxy terminus) family) also appear to modify Axin’s stability [108,109].

Axin is also known to be SUMOylated. The earliest role of SUMOylation in Axin regulation was revealed by the discovery of Axam (Axin associated molecule), an enzyme that possesses deSUMOylation activity [110,111]. Axam formed a complex with Axin and prevented its interactions with Dvl [110]. Later on, studies reported that Axin is SUMOylated and that this modification affects its role in the JNK pathway [112] and may protect Axin from ubiquitination [113].

Adding to its complex mode of regulation, Axin is proposed to utilize an autoinhibitory mechanism. The N-terminus region of the protein was earlier suggested to play an inhibitory role in binding to its partner proteins [7,114] and later shown to associate with the C-terminus, thus forming a closed conformation during the WNT signaling-off state [115]. Thus, Axin can adopt different conformational states depending on its assembly with the destruction complex or the “WNT-LRP5/6 signalosome” (Reviewed in [93]).

Yet another mechanism by which Axin function is modulated involves miRNA (microRNA)-mediated gene silencing. Experiments in the *Drosophila* system showed that *Axin* was negatively regulated by *miR-315* via conserved 3′- UTR (untranslated region) miRNA consensus sequences [116]. Likewise, studies using different human cell types demonstrated that the *Axin2* transcript was targeted by *let-7f* at the 3′ UTR, by *hsa-miR-34a* at both the 5′ UTR and 3′ UTR, and by *miR-205* at the 3′ UTR, which regulated expression of a WNT/β-catenin target gene and a β-catenin-activated reporter [117,118,119].

The findings summarized above show that Axin is subjected to multiple modes of regulation. Although the full picture of its regulatory mechanism is far from complete, it is evident that alterations help modulate Axin’s function and its interactions with other cellular factors and signaling pathway components.

## 5. Regulation of Developmental Processes in *C. elegans*

As mentioned above, the *C. elegans* genome encodes two Axin family members, PRY-1 and AXL-1. Although both proteins act as scaffolds to recruit other factors, major domains (RGS and DIX) are not well conserved (Figure 4). Of the two, PRY-1 has been investigated in some detail. The protein shows an overall 18–21% amino acid similarity with vertebrate and D-Axin. This level of conservation is primarily restricted to the RGS and DIX domain, with 27% identity (48% similarity) and 31% identity (49% similarity) with the respective domains of D-Axin [120]. Apart from the RGS and DIX domains, PRY-1 has no obvious GSK-3β- and β-catenin binding region(s). Despite this sequence discrepancy, genetic and biochemical experiments showed that PRY-1 acts as a scaffold for components of the destruction complex and negatively regulates canonical WNT signaling [120].

The genetic epistasis experiments confirmed that, similar to mammalian Axin, *pry-1* functions upstream of bar-1 (beta-catenin/armadillo related)/β-catenin and pop-1 (posterior pharynx defect 1)/TCF/LEF and downstream of egl-20 (egg laying defective 20)/WNT and mig-5 (abnormal cell migration 5)/Dvl, thus establishing it as a core component of the canonical WNT signaling pathway in *C. elegans* (reviewed in [121]). Additionally, when introduced in vertebrates, PRY-1 behaves as a functional Axin homolog, as its overexpression in zebrafish rescued the phenotype of Axin-mutation, masterblind, and inhibited WNT signaling in mammalian cells based on a TCF reporter analysis [120]. Consistent with its involvement in many processes, *pry-1* is broadly expressed during development, starting from embryogenesis [120]. At the early L1 stage, *pry-1* is mainly localized to the Q neuroblast cells (QL and QR), seam cells (V5 and V6), ventral hypodermal (P) cells (P7/8 to P11/12), body-wall muscle cells, and neurons in the head, tail, and ventral nerve cord. In addition, *pry-1* continues to be expressed in all seam cells and QL/R cells through the late-L1 stage. At later stages, *pry-1* expression persists in hypodermal cells and several neurons in the ventral cord, head, and tail ganglia [120]. Furthermore, *pry-1* expression is also observed in reproductive tissues, including vulval precursors and their progeny, as well as the male tail. A similar expression pattern of *pry-1* ortholog, *Cbr-pry-1*, was also seen in Caenorhabditis briggsae, a sister species of *C. elegans* [122]. In agreement with these expression data, constitutive activation of WNT signaling (due to the loss of PRY-1 function) causes a wide range of defects in *C. elegans* that are discussed below.

The other *C. elegans* Axin-like protein, AXL-1, also acts as a functional ortholog of Axin to regulate the canonical WNT signaling [21]. The protein shows an overall 14–16% identity to members of the D-Axin and vertebrate Axin1 and Axin2, and 20% identity to PRY-1. Similar to PRY-1, sequence conservation is restricted to the RGS and DIX domains (24% and 35%, respectively), with no obvious domains for GSK-3β and β-catenin binding [21] (Figure 4). Functional studies revealed that AXL-1 physically interacts with GSK-3/GSK-3β, MIG-5, and DSH-2 (dishevelled related 2)/Dvl, but not APR-1 (APC related 1)/APC to form a destruction complex with BAR-1 [21]. This partial destruction complex is predicted to enable BAR-1 phosphorylation by GSK-3 to inhibit WNT signaling. Furthermore, similar to PRY-1, AXL-1 overexpression inhibited WNT-induced TCF reporter in mammalian cells, suggesting its functional interaction with mammalian GSK-3β and β-catenin.

Although both AXL-1 and PRY-1 are components of the WNT signaling pathway, they are not functionally interchangeable and perform partially overlapping roles in downregulating BAR-1 signaling in developmental processes (see below) [21]. In addition, AXL-1 functions independently of PRY-1 in axonal migration and excretory cell development. Recently, Chen et al. [123] reported a novel role for AXL-1 in aging. The authors showed that following metformin treatment, AXL-1 localizes to lysosomes and regulates the PAR-4 (abnormal embryonic partitioning of cytoplasm 4)/LKB1-dependent lysosome pathway and subsequently activates AAK-2 (AMP activated kinase 2)/AMPK to extend the lifespan of *C. elegans* [123].

Below, we describe the major developmental events and tissues that depend on PRY-1 and AXL-1 function and their interactions with other cellular factors.

### 5.1. Neuronal Development

As mentioned earlier, Axin’s role in mice was initially discovered based on characterization of fused locus [1]. In the absence of Axin function, mice exhibited neurological defects. Additionally, the animals showed a neuroectodermal phenotype, which included either incomplete closure or malformation of the head. Since then, Axin family members in other organisms have been found to be essential for neuronal development. In *C. elegans*, *pry-1* mutants exhibit defects in some of their neurons (reviewed in [121]). These include the Q neuroblast system, which consists of a pair of cells, i.e., the QL cell (left lateral side) and QR cell (right lateral side), in the animal (Figure 5A). Interestingly, while the lineages of QL and QR cells are identical, both cells and their descendants migrate in opposite directions, i.e., anterior in the case of QR and posterior in the case of QL (Figure 5A). The progeny of these two neuroblasts give rise to different types of neurons during larval development.

Molecular genetic studies have shown that *pry-1*-mediated WNT signaling is essential for the migration of Q-lineage cells and guiding them along specific trajectories [126]. PRY-1 activity is specifically needed in the QR cell to restrict *mab-5 (male abnormal 5)/Hox (homeobox)* expression and to enable anterior migration of their descendants. Loss of PRY-1 function mimics constitutively active WNT pathways with high MAB-5 expression in the QR cell and their progeny, resulting in their migration in an opposite (posterior) direction [120] (Figure 5A,B). Genetic studies have identified other components of the WNT pathway including the ligand, EGL-20, as well as BAR-1 [126] (Figure 5B).

Other neuronal processes in which roles for *pry-1* have been demonstrated include axon guidance and synapse formation. Axonal function was uncovered in a genetic screen using an RNAi (RNA interference)-hypersensitive strain [127]. It was found that *pry-1* RNAi caused defects in ventral cord neurons, such as branched commissures and abnormal midline crossing. In a separate study, Schneider et al. [128] reported that *pry-1* acts in a canonical WNT-β-catenin pathway to promote synapse formation of a specific motor neuron, based on results showing that *pry-1* mutants enhanced movement defects in animals lacking *unc-4 (uncoordinated 4)/Hox* function.

In addition to its essential function in the development of neurons, *pry-1* may also participate in neuroprotection in adults. This possibility is supported by our findings that *pry-1* mutants show accelerated degeneration of dopaminergic neurons (S. Taylor, unpublished) (Figure 5C). Whether such a role of *pry-1* involves other components of the WNT signaling pathway remains to be investigated. In this regard, it is worth mentioning that WNT signaling has been linked to neurodegenerative diseases, such as Alzheimer’s and Parkinson’s (reviewed in [129,130]). One of the ways whereby *pry-1*-mediated WNT signaling may protect neurons in *C. elegans* is by regulating the expression of genes that confer neuroprotection. This hypothesis is based on our preliminary observation that the transcription of *manf-1*, a homolog of mammalian MANF (Mesencephalic astrocyte-derived neurotrophic factor) [131], was significantly downregulated in *pry-1* mutant worms (S. Taylor, unpublished) (Figure 5D). In the future, it will be interesting to investigate *pry-1*′s role in *manf-1* regulation and its link to neuroprotection.

### 5.2. Embryogenesis

Embryogenesis in *C. elegans* is another process that depends on the *pry-1*-mediated, non-canonical WNT signaling pathway. This divergent pathway, known as the WNT-β-catenin-asymmetry pathway, has been shown to control the division of several different types of somatic cell, such as EMS blastomeres in the embryo and larval seam cells (reviewed in [121]). During early embryonic development, the zygote divides into a large anterior blastomere (AB) and a small posterior blastomere (P1) (Figure 6A). P1 then divides to give rise to EMS and P2 blastomeres. The division of EMS has been studied in some detail, which is regulated by the WNT-asymmetry pathway. Upon receiving the WNT ligand, MOM-2, from the adjacent P2 blastomere, the EMS divides to give rise to MS (producing mesoderm) and E (producing endoderm) blastomeres with different cell fates [132,133] (Figure 6A). While cells of the MS lineage contribute to mesodermal tissues (i.e., pharynx and muscles), cells of the E lineage generate endodermal tissues (i.e., intestine).

In the event of asymmetric division, WNT pathway components are asymmetrically localized with MOM-5 (more of MS 5)/Frizzled (Fz), DSH-2, and MIG-5 in the posterior cortex [134,135], and with WRM-1, APR-1, PRY-1, and LIT-1 in the anterior cortex [136,137]. Subsequently, during telophase, WRM-1, SYS-1 (symmetrical sister cell hermaphrodite gonad defect 1)/β-catenin, and LIT-1 preferentially localize to the posterior nucleus [136,137,138,139,140], whereas POP-1 is found mostly in the anterior nucleus (POP-1 asymmetry) [141,142] (Figure 6A). Consistent with its localization in the anterior cortex, PRY-1 antagonizes WRM-1 function, leading to low WRM-1 activity in anterior nucleus that helps establish POP-1 asymmetry [136].

### 5.3. Seam Cell Development

Similar to the EMS, the PRY-1-mediated WNT-asymmetry pathway is also essential for seam cell development (reviewed in [121]) (Figure 6B). Seam cells are lateral hypodermal cells that give rise to specialized adult cuticular structures, namely the alae (Reviewed in [143]). During early development, a newly hatched L1 stage worm possesses 10 seam cells on either side of the body along the anterior posterior axis [144]. These cells undergo stage-specific divisions (mostly asymmetric) to produce anterior daughters with hypodermal fates and posterior daughters with seam cell fates, and ultimately differentiate to form alae by the end of the L4 stage. The components of WNT-asymmetry pathway in this developmental system are the same as those involved in EMS divisions (Figure 6A, B). Thus, prior to their division, APR-1 and PRY-1 localize to the anterior cortex of a seam cell, whereas MOM-5, DSH-2, and MIG-5 are found in the posterior cortex [134,145,146,147]. The fates of daughter cells depend on the nuclear levels of SYS-1 and POP-1, such that the anterior nucleus possesses high POP-1 and low SYS-1, and the posterior nucleus possesses low POP-1 and high SYS-1 (Figure 6B).

In support of its role in the asymmetric division of seam cells, *pry-1* mutation disrupts the nuclear localization of WRM-1, SYS-1, and POP-1 [66,134,149], thereby leading to increased seam cell proliferation [65,66]. Work from our lab has shown that in the absence of PRY-1 function, POP-1 localization is disrupted in seam cell daughters [66]. As expected, RNAi knockdown of *wrm-1* and *lit-1* suppressed the seam cell phenotype in *pry-1* mutants whereas *pop-1* RNAi exacerbated the defect.

The temporal division pattern of seam cells relies on several heterochronic miRNAs and their target genes (reviewed in [150,151]. We analyzed the miRNA transcriptome in *pry-1* mutants, which revealed five DE (differentially expressed) miRNAs of *lin-4 (lineage defective 4)* and *let-7 (lethal 7)* families. Further experiments revealed that all DE miRNAs were repressed by PRY-1 in a POP-1-dependent manner [66] (Figure 6C).

In addition, *pry-1* plays a role in V-lineage development in males, where it restricts expression of the Hox gene *mab-5* to posterior V-cell descendants, which ensures correct specification of cell fates and leads to the formation of sensory rays, alae, and the postdeirid [148]. Thus, in males, only the V5 and V6 cell lineages (expressing *mab-5*) generate rays whereas the V1–V4 lineages (with no *mab-5* expression) give rise to alae [152,153]. Males with no *pry-1* function show defective alae and ectopic rays as the V cells can now differentiate to make more rays due to the inappropriate expression of *mab-5* [148] (Figure 6D). Moreover, altered *mab-5* expression in *pry-1* mutants inhibits the formation of the postdeirid, a sensory structure resulting from the differentiation of V5.pa descendants [148].

### 5.4. Vulva Development

Although *pry-1* was identified initially based on its role in the Q neuroblast-cell lineage, *pry-1*-mutant animals were subsequently reported to exhibit defects in vulva formation (Figure 7A). *C. elegans* vulva has been studied extensively to understand how signal transduction pathways control cell fates and organogenesis (reviewed in [154]). As a reproductive organ, the vulva serves as a system for mating with males and egg laying. The organ develops from three of the six equipotential groups of P-lineage cells (Pn.p, n = 3–6), termed vulval precursor cells (VPCs), which are induced to adopt primary (1^0^ - P6.p) and secondary (2^0^ - P5.p and P7.p) cell fates. Mutations in *pry-1* cause more than three VPCs to get induced and lead to the formation of ectopic pseudo-vulvae-like structures in adults, which is referred to as the multivulva (Muv) phenotype [155] (Figure 7A,B). Genetic experiments revealed that the gene acts in the canonical WNT-β-catenin pathway to repress inappropriate induction of vulval precursors. Because of its role as a negative regulator, reduction or elimination of *pry-1* function leads to the constitutive activation of WNT signaling and the dysregulation of downstream targets. One such target is the homeobox family member, *lin-39 (lineage defective 39)*, which is necessary for *pry-1*-mediated vulval development [155].

Axin family members have also been genetically characterized in other nematodes. Our laboratory identified mutations in the *C. briggsae pry-1* ortholog (*Cbr-pry-1*) and showed that *Cbr-pry-1* functions in vulva development in a *Cbr-bar-1/β-catenin*-, *Cbr-pop-1/tcf/lef*-, and *Cbr-lin-39/hox*-dependent manner [122] (Figure 7B). Research in a more distant nematode, *Pristionchus pacificus* (Diplogastridae family), uncovered an Axin family member that is most closely related to *C. elegans axl-1* [158]. Mutations in *Ppa-axl-1* caused a Muv phenotype suggesting that the gene plays an important role in vulva development. Further experiments revealed that *Ppa-axl-1* genetically interacts with WNT-β–catenin pathway components.

Experiments aimed at understanding the mechanism of *pry-1* function have discovered crosstalk between different signaling pathways. Gleason et al. [155] found that *pry-1* mutants bypass the requirements of EGFR–Ras signaling components, which led them to propose that the WNT-β-catenin and EGFR–Ras pathways can act in parallel to promote vulva formation (Figure 7C). Subsequently, our group showed that PRY-1-mediated WNT signaling interacts with the LIN-12/Notch cascade to confer a 2^0^ fate on induced VPCs, since the loss of *pry-1* leads to inappropriate activation of the *lin-12* target gene *lip-1 (lateral-signal-induced phosphatase 1)/MAPK phosphatase* [122]. How the three pathways interact to regulate downstream targets (see a model in Figure 7C), leading to correct specification of VPC fates, is currently not understood.

### 5.5. P11/12 Development

P11 and P12 cells also depend on *pry-1*-mediated signaling for proper differentiation of their progeny. These two cell lineages contribute to the formation of the ventral nervous system [144]. Whereas their anterior daughters, i.e., P11.a and P12.a, are neuroblasts that divide to form several neurons, the posterior daughters, i.e., P11.p and P12.p, take on different fates. The P11.p fuses with the hyp7 syncytium. The P12.p divides to generate two daughters, one of which, P12.pp (posterior daughter), undergoes programmed cell death and the other, P12.pa (anterior daughter), acquires a unique hypodermal cell fate, hyp12. In *pry-1*-mutant animals, P11.p appears to adopt a P12.pa-like fate, based on the presence of two cells having the characteristics of P12.pa [157] (Figure 7D). Mutations in *bar-1* exhibit an opposite phenotype, i.e., two P11.p-like cells [156]. Since these phenotypes may arise due to cell-fate changes at the level of P11 and P12, it is likely that PRY-1–BAR-1-mediated WNT signaling plays a role in conferring the correct fates of these two P cells.

### 5.6. Male Hook Development

The role of *pry-1* in the development of the male hook has also been investigated. The hook sensillum is a copulatory structure that helps in locating the hermaphrodite vulva during mating [157]. Cell-morphology and -lineage studies have shown that the hook is formed by the progeny of P10.p and P11.p precursors [157]. These two cells are induced to adopt 1^0^ (P11.p) and 2^0^ (P10.p) fates through the actions of the WNT, EGFR-Ras, and LIN-12 pathways. The P10.p and P11.p progeny differentiate to form a functional hook that includes sensory neurons, structural cell, and support cells. It was found that *pry-1* mutants have ectopic hook-like structures due to inappropriate induction of some of the anterior Pn.p (n = 3–8) cells that normally fuse with the surrounding hypodermal syncytium [157] (Figure 7E). This phenotype was suppressed by mutations in *bar-1* and a downstream target (*mab-5*). Additional genetic experiments led the authors [157] to propose a model in which a graded WNT signal from the tail region causes maximal activation of the pathway in P11.p, leading to inhibition of PRY-1 function and specification of the 1^0^ fate. The neighboring P10.p cell receives a comparatively lower signal and thereby adopts a 2^0^ fate.

### 5.7. Lipid Metabolism

The processes described above were related to the development of cells and tissues. Recently, our group uncovered a novel role of PRY-1 that involves the regulation of lipid metabolism [86]. In *C. elegans* and other eukaryotes, fatty acids (such as triacylglycerols, phospholipids, and sphingolipids) serve as building blocks for lipids. Fatty acids are synthesized from thioesters or isoprene units via condensation reactions (Reviewed in [159]). We analyzed the mRNA transcriptome in *pry-1* mutant animals and determined that DE genes are highly linked to biological processes such as lipid metabolism, cellular responses to lipids, and aging [86]. Functional studies involving a subset of DE genes revealed that *vits* (*vitellogenins*, yolk lipoproteins) and *fats (fatty acid desaturases)* participate in *pry-1*-mediated lipid metabolism. Consistent with these findings, *pry-1* mutants exhibited defects in lipid content (Figure 8A), egg laying, and survival following starvation [86]. Genetic experiments showed that *vits* act downstream of *pry-1*, and RNAi knockdowns of *vit* genes rescued lipid defects in *pry-1* mutants.

In *C. elegans*, MUFAs and PUFAs (monounsaturated and polyunsaturated fatty acids, respectively) can be derived from saturated fatty acid precursors by the actions of fat desaturases (FAT-5, FAT-6, and FAT-7) [159,160]. The expression of desaturases is regulated by transcription factors SBP-1 (sterol regulatory element-binding protein 1, SREBP-1 family) and two HNF4 (hepatocyte nuclear factor 4) families of nuclear receptors NHR-49 (nuclear hormone receptor 49) and NHR-80 (nuclear hormone receptor 80) (Reviewed in [159]). More recently, SBP-1-mediated regulation of MUFA synthesis was shown to extend lifespan when exposed to anti-aging drug combinations [161]. We found that *pry-1* mutants had reduced transcription of fat desaturases and *sbp-1*. However, transcription of the *nhr-49* and *nhr-80* genes was unaltered. Thus, a working model is that *pry-1* acts via *sbp-1* to regulate the expression of *fat* genes that, in turn, regulate lipid synthesis [86]. This model (Figure 8B) is supported by the findings that fatty acid levels were reduced in *pry-1* mutants and supplementing the diet with oleic acid, a MUFA, rescued lipid defects in mutant animals [86].

The requirements of lipids in many biological processes and across multicellular eukaryotes is well documented. These macronutrients are vital for proper growth and reproduction; however, their excessive intake contributes to a variety of diseases in humans. Changes in lipid metabolism could affect a host of processes including signaling, cell structure, and aging [162]. Outside of the *C. elegans* system, Axin family members are reported to function in lipid biology. Axin expression in mice contributes to an age-related increase in adiposity in thymic stromal cells [59]. A recent study showed that Axin knockdown in mouse liver impaired AMPK activation and abrogated AMPK-LKB1 colocalization upon starvation [58]. Therefore, understanding the role of PRY-1 and its homologs in maintaining energy homeostasis is of considerable interest in biomedical research.

## 6. Major Findings from *C. elegans* Studies

As described in the previous section, PRY-1 and AXL-1 show significant sequence divergence in GSK-3β and β-catenin binding domains. In spite of this, both of these proteins physically interact with BAR-1 and regulate WNT-β-catenin signaling. The functional conservation may be brought about by conserved structures of RGS and DIX domains, thereby making them bona fide Axin family members.

Similar to other eukaryotes, *C. elegans* Axins participate in developmental processes such as cell proliferation, cell differentiation and cell migration. The examples include the formation of the vulva and male hook, P11/12 fate specification, and neuronal development. All of these involve interactions with WNT-β-catenin pathway components, suggesting that this might be the most ancestral mechanism of Axin function. Among other roles of PRY-1, its involvement in lipid metabolism is a recent discovery that may potentially fit into a broader role of Axin as an energy sensor and regulator in eukaryotes.

In addition to their conserved roles, studies in *C. elegans* have also uncovered a unique Axin-mediated WNT signaling during embryogenesis and seam cell development. Typically, interactions between β-catenin and TCF lead to transcriptional regulation of the target genes. However, this divergent WNT signaling has evolved to utilize WRM-1/β-catenin to regulate nuclear localization of POP-1/TCF, leading to the specification of asymmetric cell fates. Thus, research in the *C. elegans* model has shed a new light on the novel mechanism of Axin function in eukaryotes.

## 7. Concluding Remarks

Axin is a well-characterized scaffold protein that is conserved in eukaryotes. While vertebrate genomes carry two Axin genes, a single ancestral family member is found in invertebrates. Axin’s function has been studied mainly in the context of WNT-β-catenin signaling; however, the protein also interacts with multiple factors belonging to WNT-independent signal transduction pathways. Over the years, studies have shown that Axin homologs are necessary for the development of various tissues and cell types. In *C. elegans*, Axin’s role has been investigated during embryogenesis and larval development. Given the extensive conservation of genes and signaling mechanisms in *C. elegans*, future studies on PRY-1 and AXL-1 hold the potential to further advance our understanding of shared mechanisms of Axin function in animal development. By harnessing the power of *C. elegans* genetics, new and novel interacting partners of Axin could be identified. Additionally, the discovery of Axin-target genes could help unravel pathways and process-specific functions associated with this family of proteins.

## Figures and Tables

**Figure 1 jdb-07-00020-f001:**
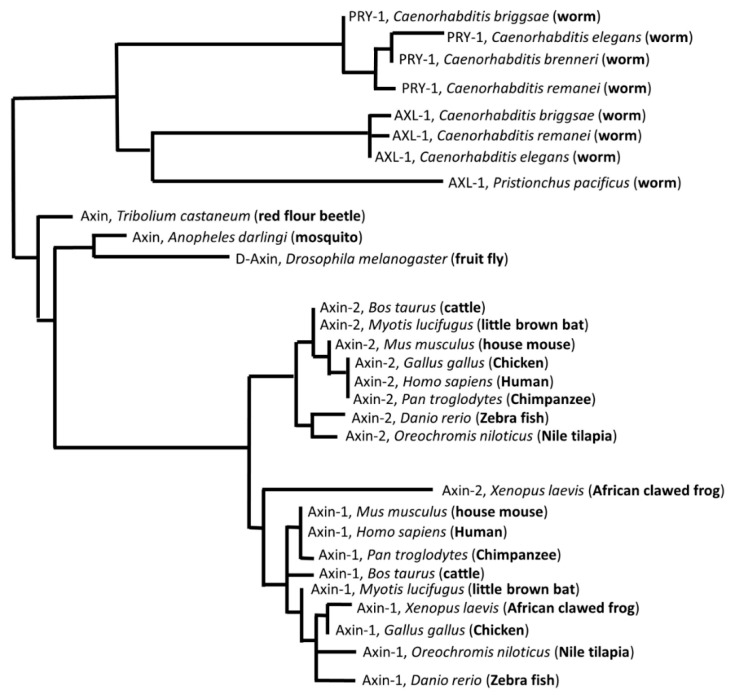
Axin family is conserved in multicellular eukaryotes. Multiple sequence alignment dendrogram was generated by LIRMM (http://www.phylogeny.fr/simple_phylogeny.cgi) using default program parameters.

**Figure 2 jdb-07-00020-f002:**
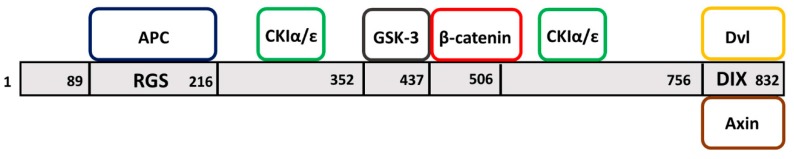
Axin binds to a wide array of proteins. Shown here are the WNT signaling components—APC, CKI, GSK-3, β-catenin and Dvl, with known relative binding positions in Mouse Axin1 (NP_001153070.1). The C-terminus domain (DIX) facilitates formation of homo and hetero dimers.

**Figure 3 jdb-07-00020-f003:**
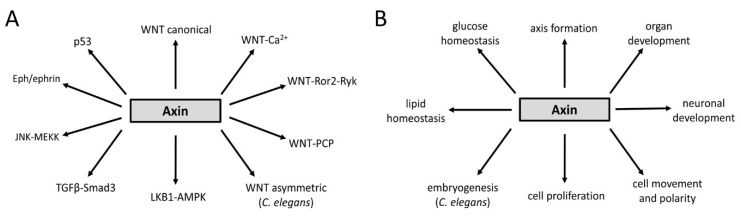
An overview of Axin’s involvement in multiple pathways (**A**) and processes (**B**), as described in this review.

**Figure 4 jdb-07-00020-f004:**
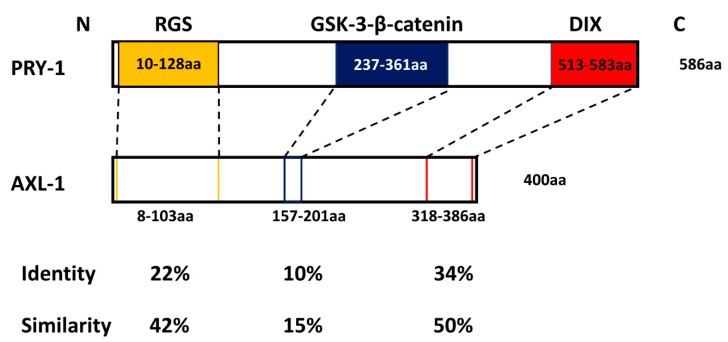
Protein sequence alignment of PRY-1 and AXL-1 in *C. elegans*. The three major domains (RGS, GSK-3-β-catenin, and DIX) in PRY-1 are indicated by colored boxes. The corresponding regions in AXL-1 and their amino acid sequence identity and similarity are also shown. Sequence alignment was done using CLUSTAL W and T-COFFEE (http://www.clustal.org/clustal2/, http://tcoffee.crg.cat) [124,125].

**Figure 5 jdb-07-00020-f005:**
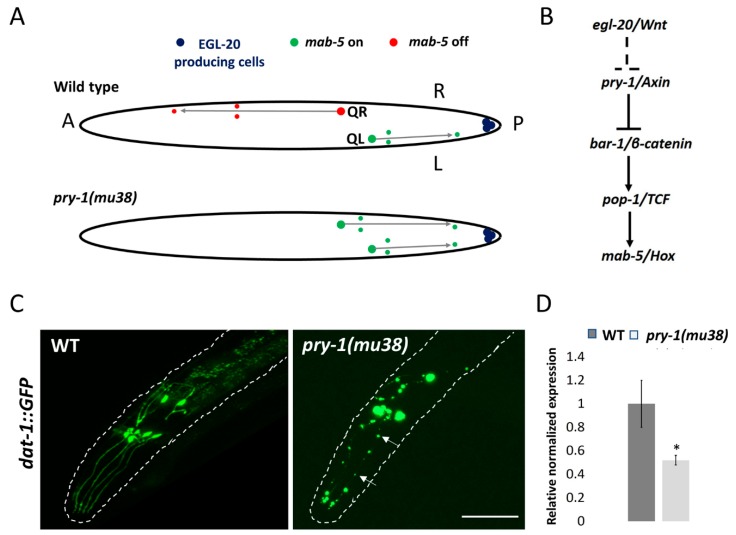
PRY-1 regulates neuronal development in *C. elegans*. (**A**) EGL-20/WNT signaling activates the Hox gene *mab-5* in QL to induce posterior migration of QL descendants. *mab-5* is not activated in QR, and as a consequence, the QR descendants migrate in the default anterior direction. In *pry-1(mu38)* mutant animals, *mab-5* is ectopically expressed in QR leading to the migration of QR descendants towards posterior region. (**B**) PRY-1 acts in the canonical WNT signaling to regulate the expression of *mab-5/ Hox* target gene. The dotted line indicates indirect interaction. (**C**) *pry-1* mutants exhibit defects in dopaminergic neurons (marked with *dat-1p::GFP*). The cell bodies are frequently missing or appear abnormal and dendrites show punctate-like patterns (arrows) (scale bar represents 0.05 µm). (**D**) qPCR experiment shows that *manf-1* is significantly downregulated in *pry-1* mutant adults (* *p* < 0.05, two batches).

**Figure 6 jdb-07-00020-f006:**
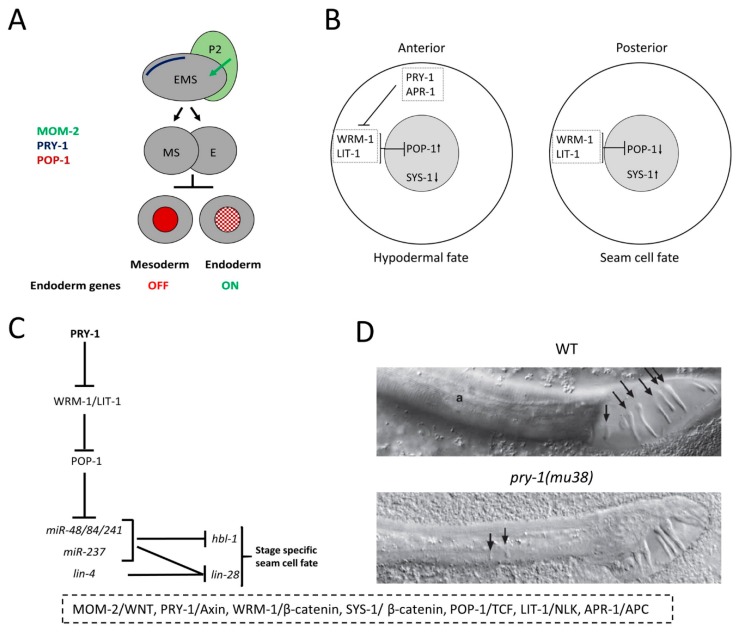
PRY-1 negatively regulates asymmetric cell division during *C. elegans* development. (**A**) Model for EMS division. PRY-1, located in the anterior cortex of EMS during asymmetric division, is involved in conferring endodermal and mesodermal fates of daughter cells. (**B**) Similar to EMS division, a model for seam cell division. PRY-1 negatively regulates WNT signaling in the anterior cell, which ultimately adopts a hypodermal fate. (**C**) A genetic pathway consisting of PRY-1-mediated regulation of heterochronic miRNAs and their targets during seam cell development. (**D**) *pry-1(mu38)* males show defective tail morphology. In wild-type (WT) animals, rays are located in the fan-like region (marked by arrows in the upper panel). In *pry-1* mutants, alae have been replaced with ectopic rays (arrows in the lower panel). The vertebrate homologs of *C. elegans* genes are listed on the bottom. Panels B and C adopted with permission from [66] and panel D from [148].

**Figure 7 jdb-07-00020-f007:**
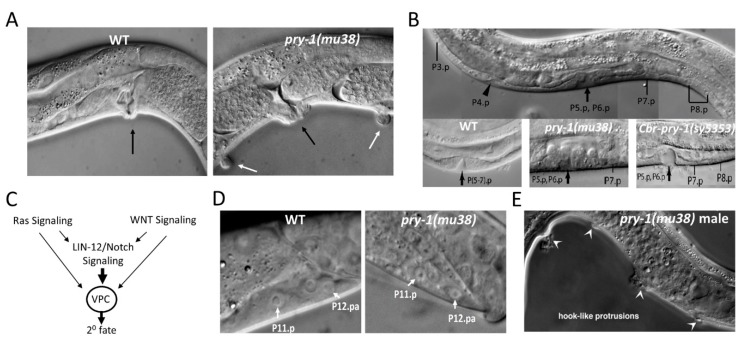
*pry-1* is necessary for the development of P lineage cells. (**A**) Multiple vulva-like protrusions seen in a *pry-1(mu38)* animal (white arrows). Black arrows mark the main vulva. WT, wild-type (**B**) VPC fates are defective in *pry-1(mu38)* animals. Unlike the wild-type, where the progeny of P(5-7).p give rise to the vulva, P7.p in *pry-1(mu38)* and P7.p and P8.p in *Cbr-pry-1(sy5353)* animals remain unfused (panel adapted with permission from [122]). (**C**) A proposed model of interactions between WNT, Ras, and Notch pathways to specify the 2^0^ fate of induced VPCs. (**D**) *pry-1(mu38)* mutants show an extra P12.pa-like cell in the place of P11.p. The wild-type P11.p has a large nucleus and a nucleolus compared to P12.pa, which is noticeably smaller. In the case of *bar-1(ga80)* mutants, an opposite phenotype is seen, i.e., two P11.p-like cells [156]. (**E**) Ectopic hook-like structures in *pry-1* mutant males are marked by arrowheads (panel adapted with permission from [157]).

**Figure 8 jdb-07-00020-f008:**
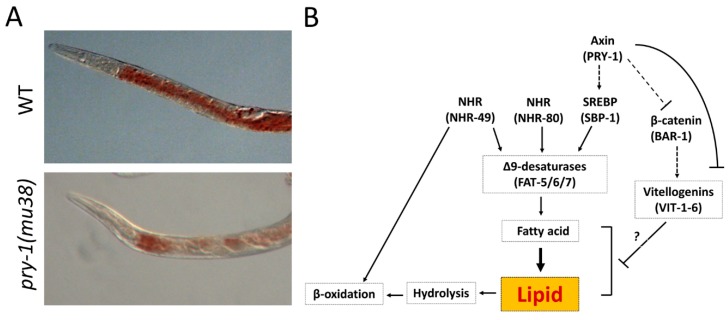
*pry-1* is necessary to regulate lipid metabolism. (**A**) *pry-1* mutant animals show reduced lipid content compared to wild-type (WT) animals as revealed by Oil Red O staining. (**B**) A model for *pry-1* genetic network in regulating lipid synthesis based on findings presented in [86]. *pry-1* acts upstream of *sbp-1* and in parallel to *nhr-49* and *nhr-80*. *pry-1* also regulates the expression of *vit* genes. Whether *bar-1* participates in this process (dotted connecting lines) and how *vits* affect lipid synthesis (question mark) are currently not understood.

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
