# Peer review of "Axin Family of Scaffolding Proteins in Development: Lessons from C. elegans"

_jdb, 2019, doi:10.3390/jdb7040020_

Round 1
Reviewer 1 Report
This review by Mallick et al describes the different activities of Axin in Development with a focus on C. elegans studies. Overall the review is interesting and well written however there are major concerns that need to be address before publication.
The review is organized in two main parts. The first part provides a background on Axin activities in all species and is subdivided in 3 sections: 1-Protein structure; 2- Developmental roles; 3- Axin activities in signaling pathways; 4-Regulatory mechanisms of Axin activities. The second part focuses on C. elegans studies.
My major concern is the readability of the review.
I strongly advise to fully separate studies of Axin in c. elegans versus in other species: lanes 126 to 174 should be in the c elegans part.
A better discussion is necessary: 1 to understand the connection between part 1 and 2, 2- to discuss what is conserved and what is not conserved between Axin roles in different pathways, molecular activities and functions 3- to which extend c.elegans studies bring a new light on Axin proteins functions understanding.
For example I would have liked to understand why PRY1 and AXL1 do not carry GSK3ß and ß-Catenin binding domains whereas they appear to functionally behave as their vertebrates orthologs in regulating the Wnt canonical pathway.
The authors should be more precise in their assertion of Wnt activities and provide what is the read out (for example lanes 165-166 and others parts of the manuscript).
Axin is interacting with a large panel of proteins. If this review intends to reach a large readership couple of modifications ought to be made.
1- I suggest to provide at least one and maybe two schemes summarizing the different activities of Axin and/or the different types of interactions between Axin and signaling pathways. These schemes could complement Figure 2, which is not very instructive.
2- when possible always provides the name in c-elegans and the name of the ortholog in vertebrates as in Figure 7.
The review mentions briefly the different activities of Axin1 and 2. The authors should be more precise on which activities of Axin are generic, or specific of Axin 1 or Axin 2 and whether such specificities are conserved between species.
I am not certain that unpublished work is acceptable in JDB lanes 355, 363 etc.
Importantly Axin is a scaffold protein. After reading this review it is not clear to me how this scaffolding function influences Axin biological activities (Lanes 83-84), which would be useful specifically considering the first sentence of the abstract.
Minor comments
1-Figure 6C is too small.
2-What does enrichment of pathways means (lane 104)
3-In the Invertebrates part the formatting is not respected
Author Response
Response to reviewer comments
We are encouraged by constructive comments of both reviewers. Below we provide a detailed point-by-point response to issues that have been raised. In almost all cases, our explanations are supported by appropriate modifications in the manuscript (identified by track changes). These include two new figures and a brand new sub-section. Overall, these changes have resulted in improved quality of the review article, for which we sincerely thank the reviewers. We are confident that the revised version will meet their expectations.
Reviewer 1
This review by Mallick et al describes the different activities of Axin in Development with a focus on C. elegans studies. Overall the review is interesting and well written however there are major concerns that need to be address before publication.
The review is organized in two main parts. The first part provides a background on Axin activities in all species and is subdivided in 3 sections: 1-Protein structure; 2- Developmental roles; 3- Axin activities in signaling pathways; 4-Regulatory mechanisms of Axin activities. The second part focuses on C. elegans studies.
My major concern is the readability of the review.
I strongly advise to fully separate studies of Axin in c. elegans versus in other species: lanes 126 to 174 should be in the c elegans part.
We have made the changes as suggested.
A better discussion is necessary: 1 to understand the connection between part 1 and 2, 2 to discuss what is conserved and what is not conserved between Axin roles in different pathways, molecular activities and functions, 3 to which extend c.elegans studies bring a new light on Axin proteins functions understanding. For example I would have liked to understand why PRY1 and AXL1 do not carry GSK3ß and ß-Catenin binding domains whereas they appear to functionally behave as their vertebrates orthologs in regulating the Wnt canonical pathway.
The following changes have been made to address the concerns.
Added a new paragraph (lines 136-140). Modified the last paragraph in section ‘Axin proteins interact with many factors including signaling pathway components' (lines 332-337). Wrote a brand new sub-section 'Major findings from elegans studies’ (lines 725-742)
The authors should be more precise in their assertion of Wnt activities and provide what is the read out (for example lanes 165-166 and others parts of the manuscript).
We have resolved this issue. For example, see lines 453 that explicitly mentions ‘Wnt-induced TCF reporter’. Similar changes have also been made elsewhere in the manuscript.
Axin is interacting with a large panel of proteins. If this review intends to reach a large readership couple of modifications ought to be made.
1- I suggest to provide at least one and maybe two schemes summarizing the different activities of Axin and/or the different types of interactions between Axin and signaling pathways. These schemes could complement Figure 2, which is not very instructive.
We have now added two new diagrams (see panels A, B in Figure 3).
2- when possible always provides the name in c-elegans and the name of the ortholog in vertebrates as in Figure 7.
This has been addressed. See the revised figures 5 and 6.
The review mentions briefly the different activities of Axin1 and 2. The authors should be more precise on which activities of Axin are generic, or specific of Axin 1 or Axin 2 and whether such specificities are conserved between species.
Most studies on vertebrate Axin family have focused on Axin1 (commonly referred as Axin in our article). We have clarified this point in the revised manuscript (lines 39-44)
I am not certain that unpublished work is acceptable in JDB lanes 355, 363 etc.
We don’t see any issue with this. Unpublished data have been cited in other similar review articles in the same issue of Journal of Developmental Biology.
Importantly Axin is a scaffold protein. After reading this review it is not clear to me how this scaffolding function influences Axin biological activities (Lanes 83-84), which would be useful specifically considering the first sentence of the abstract.
We have written a new paragraph (lines 136-140) to address this issue.
Minor comments
1-Figure 6C is too small
We have made changes in the figure panel.
2-What does enrichment of pathways means (lane 104)
The sentence has been modified to make the point clearer (lines 110-112).
3-In the Invertebrates part the formatting is not respected
If this comment refers to gene/protein names, then appropriate changes have been made to ensure that the names are consistent with the vertebrate naming system. It is important to point out that according to C. elegans nomenclature, gene names are written in small letters and italicized whereas protein names are in capital letters without any italics.
Reviewer 2 Report
The review, “Axin family of scaffolding proteins in development: Lessons from C. elegans,” is a very thorough and historical account of the study of the Axin genes. I found the beginning and the final sections particularly interesting.
The description of early Axin studies is very informative. The authors even tackle the thorny issue of Axin phosphorylation, unraveling the many early findings that suggested both positive and negative regulation.
The authors should also include a description of how the Wnt pathway is different in C. elegans from the vertebrate pathway as this will help with the explanations for the last section. For example, how is the relationship between Pop-1 and Wrm-1 different from b-catenin and TCF.
Lines 176 to 251. The middle section, describing the interaction of Axin with other pathways and the many binding factors discovered is the least interesting. There are a lot of one-off experiments and controversial findings which are not really necessary for this review.
The C. elegans section is excellent. I would possibly add that lipid metabolism and SBP-1/SREBP regulation in C. elegans aging was recently shown to be activated downstream of anti-aging drugs, Admasu et al. Dev Cell. 2018.
Minor:
Line 38—“…both Axin proteins activate…”. Axin can play both positive and negative roles in Wnt signaling so please change activate to regulate.
Figure 2—This figure would benefit from color. Also, b-catenin isn’t in a box.
Line 105—Hippo pathway?
Line 147—Neuroblast, not blasts
Line 527—a MUFA, not an MUFA
Author Response
Response to reviewer comments
We are encouraged by constructive comments of both reviewers. Below we provide a detailed point-by-point response to issues that have been raised. In almost all cases, our explanations are supported by appropriate modifications in the manuscript (identified by track changes). These include two new figures and a brand new sub-section. Overall, these changes have resulted in improved quality of the review article, for which we sincerely thank the reviewers. We are confident that the revised version will meet their expectations.
Reviewer 2
The review, “Axin family of scaffolding proteins in development: Lessons from C. elegans,” is a very thorough and historical account of the study of the Axin genes. I found the beginning and the final sections particularly interesting. The description of early Axin studies is very informative. The authors even tackle the thorny issue of Axin phosphorylation, unraveling the many early findings that suggested both positive and negative regulation.
The authors should also include a description of how the Wnt pathway is different in C. elegans from the vertebrate pathway as this will help with the explanations for the last section. For example, how is the relationship between Pop-1 and Wrm-1 different from b-catenin and TCF.
This has been addressed in a separate sub-section “Major findings from C. elegans studies (lines 725-742)."
Lines 176 to 251. The middle section, describing the interaction of Axin with other pathways and the many binding factors discovered is the least interesting. There are a lot of one-off experiments and controversial findings which are not really necessary for this review.
Our article is aimed at readers seeking to gain a comprehensive understanding of the Axin family in biological processes. Therefore, it is desirable to report both major and minor findings on these proteins. We have avoided a few one-off experiments that did not involve Axin’s role in well-studied pathways and processes. The current manuscript includes data that is relevant and up to date.
The C. elegans section is excellent. I would possibly add that lipid metabolism and SBP-1/SREBP regulation in C. elegans aging was recently shown to be activated downstream of anti-aging drugs, Admasu et al. Dev Cell. 2018.
We have added the reference as suggested.
Minor:
Line 38—“…both Axin proteins activate…”. Axin can play both positive and negative roles in Wnt signaling so please change activate to regulate.
This has been corrected.
Figure 2—This figure would benefit from color. Also, b-catenin isn’t in a box.
This has been done as suggested.
Line 105—Hippo pathway?
Authors of the paper who reported the findings have stopped short of calling it a Hippo pathway. The study focused on genes that are affected by YAP. We have revised the sentence to make it clearer (lines 110-112).
Line 147—Neuroblast, not blasts
This has been corrected.
Line 527—a MUFA, not an MUFA
This has been corrected.
Round 2
Reviewer 1 Report
The revised version of the review by Mallick et has been majorly improved especially the readability and is now of very good quality.
I still have small remarks detailed below.
Lane 132-133 it should be the roles of both these family members are discussed in a separate section.
Lanes 197-198 and processes (B) as described in this review.
I personally thinks that considering the title of the review the Part “Major findings from C.elegans studies” should be integrated into the Conclusive remarks.
